# Exploring EFL teachers' beliefs and practices of formative assessment in Chinese context

Wanrong Lei ⓘ *, Zhibin Lei

School of Foreign Languages, Hunan First Normal University, Changsha, China

* leiwanrong001@126.com

## Abstract

Formative assessment (FA) has become central to effective EFL pedagogy, promoting feedback, engagement, and self-regulated learning. Yet in the Chinese EFL context, little is known about how university teachers actually conceptualize and enact FA in their classrooms. This study explores EFL teachers' beliefs, self-reported practices, and challenges in implementing FA through a mixed-methods design that combines a survey (N = 288) with follow-up interviews (n = 11). Results revealed that most teachers hold positive beliefs about FA and acknowledge its potential to enhance student motivation and engagement. However, the frequency of FA practices varies significantly among the participants. Teachers tend to select their FA strategies within institutional limits and rely on technological platforms for implementing these strategies. Despite acknowledging FA's role in informing teaching improvements and student learning, it was frequently used for grading and managing classroom participation. Challenges included time constraints, large class sizes, inadequate technological support, difficulty in establishing clear assessment criteria, and underutilization of self- and peer assessment. These findings highlight the necessity for multifaceted interventions to better align EFL teachers' beliefs with their practices in FA.

## 1. Introduction

Formative assessment (FA) has garnered significant attention in educational research and practice over the past few decades as a vital approach to enhancing teaching and learning outcomes. Unlike summative assessment, which evaluates learning at the end of an instructional period, formative assessment focuses on monitoring student progress and providing ongoing feedback to improve the teaching and learning process [1].

In the Chinese context of English as a Foreign Language (EFL), formative assessment has been a focus in English education. It has also been mandated into college English teaching for over two decades by the Ministry of Education [2]. However, its implementation faces challenges due to teachers' limited assessment literacy and

**Data availability statement:** All relevant data are within the manuscript and its Supporting information files.

**Funding:** This work was supported by the Hunan Provincial Department of Education (Grant No. HNJG-20231361; recipient: W.L.). The funders had no role in the study design, data collection and analysis, decision to publish, or preparation of the manuscript.

**Competing interests:** The authors have declared that no competing interests exist.

the exam-oriented culture in Chinese education [3,4]. These challenges create a gap between what education policies require and what happens in classrooms.

Recent studies highlight several factors influencing teachers in successfully using formative assessment. A study revealed that the effectiveness of formative assessment relies on teachers' understanding and ability to its execution [5]. Moreover, integrating technology into formative assessment has introduced new opportunities and challenges for EFL teachers. Technology-enhanced formative assessment positively impacts students' language accuracy and performance [6], but it requires adequate teacher training and institutional support [7]. Additionally, teachers' professional background, institutional context, and assessment literacy can influence their beliefs and practices regarding formative assessment [8]. Therefore, there is a need for insight into these aspects to inform the design of targeted professional development programs and to build teachers' capacity to utilize formative assessment effectively [9].

However, despite substantial research that has explored formative assessment in the EFL context, a notable gap persists in understanding how Chinese university EFL teachers understand and implement formative assessment in their daily teaching [10]. The existing studies have mostly focused on teachers' assessment literacy [11] or students' perceptions [12]. However, less attention has been paid to teachers' actual formative assessment practices and the factors influencing their implementation.

To fill these gaps, this study investigates Chinese university EFL teachers' formative assessment beliefs, practices, and the challenges they face. Specifically, it seeks to answer the following research questions:

1. What are Chinese university EFL teachers' beliefs about formative assessment?

2. How do Chinese university EFL teachers implement formative assessment in their teaching practices?

3. What challenges do Chinese university EFL teachers face in implementing formative assessment?

By answering these questions, this research contributes to a deeper understanding of the complexities involved in implementing formative assessment in the Chinese EFL context, thereby enhancing teachers' capacity to utilize formative assessment by addressing their specific needs and challenges or informing teacher training programs within the EFL educational setting.

## 2. Literature Review

### 2.1 Formative assessment in EFL context

Formative assessment is defined as a classroom practice where evidence of student achievement is elicited, interpreted, and used by teachers, learners, or their peers to inform teaching and improve learning [13]. Unlike summative assessment (SA), which emphasizes measuring learning outcomes for accountability, formative assessment provides descriptive feedback during the instructional process, fostering a dynamic interplay between teaching, learning, and assessment [14]. Central to formative assessment is recognizing students as active participants, encouraging them

to set learning goals, monitor their progress, and act on feedback from teachers and peers [15,16]. These practices align closely with the principles of self-regulated learning (SRL), which is defined as a process where learners actively control their cognitive and motivational states to achieve specific goals [17]. Formative assessment thus serves as both a tool and a catalyst for developing lifelong learning skills, positioning SRL as an integral theoretical underpinning [18].

In order to translate the formative assessment principles into practice, frameworks such as the model proposed by Black and William offer practical strategies, including clarifying learning intentions, gathering learning evidence, and providing forward-focused feedback [13]. These strategies provide guidance for teachers and allow students and peers to take active roles in the assessment process, creating a collaborative learning environment [19]. Recent research has emphasized the application of these strategies in the EFL environments, where formative assessment practices such as self-assessment, peer assessment, and descriptive feedback have been shown to enhance learners' self-regulation and engagement [20,21]. However, despite these promising results, studies have indicated that EFL teachers often face challenges in fully implementing formative assessment, such as inconsistent sharing of assessment criteria, a preference for evaluative over descriptive feedback, and the insufficient use of peer- and self-assessment practices [22,23]. Additionally, students' perceptions of formative assessment influence its effectiveness: positive perceptions foster constructive learning behaviors, while negative ones may hinder engagement [24,25].

The integration of technology has expanded formative assessment's potential through tools that enhance immediacy, personalization, and interactivity. Platforms like Moodle quizzes and gamified tools (e.g., Kahoot) improve engagement, iterative learning, and summative performance [26,27]. Advanced technologies, such as AI-enabled systems, offer detailed analytics and visualizations to inform instruction. For instance, AI-powered visual report tool could enhance student self-efficacy through cognitive diagnostics and real-time feedback [28]. Despite these advancements, challenges include technological glitches, insufficient teacher training, and the labor-intensive design of technology-enhanced assessments [29]. Best practices advocate balancing automated feedback with elaborative explanations to foster both accuracy and more profound understanding [30].

The efficacy of formative assessment in the EFL context remains contingent on overcoming institutional and cultural barriers. Teachers often struggle with insufficient assessment literacy and conflicting priorities in exam-oriented environments [22]. Students' divergent attitudes towards formative assessment, especially skepticism or resistance to peer evaluation, can undermine effectiveness [24]. Additionally, technology integration, though promising, introduces new demands for teacher training and institutional support [28,29]. These challenges highlight the need for context-sensitive adaptations of formative assessment frameworks to align with local pedagogical cultures and resource constraints.

## 2.2 EFL teachers' beliefs and practices of formative assessment

Teachers' beliefs play a crucial role in shaping formative assessment practices. Research suggests that teachers who strongly believe in the benefits of formative assessment are more likely to adopt practices such as providing timely feedback, utilizing peer and self-assessment, and modifying instruction based on assessment data [24,31]. These beliefs often stem from teachers' understanding of assessment theory and their classroom experiences. For example, teachers with positive perceptions of formative assessment view it as a tool to enhance learning outcomes and student engagement, which aligns with broader educational goals [32]. However, there is often a significant dissonance between teacher beliefs and actual practice. Although teachers recognize the theoretical value of formative assessment, their ability to translate these beliefs into effective classroom practices is often limited due to situational factors such as institutional expectations and resource constraints, resulting in a tendency for their practices to be product-oriented rather than process-oriented [33,34]. This disconnect emphasizes the importance of examining how beliefs interact with other variables to influence the implementation of formative assessment.

In the context of EFL education, teachers often engage in activities such as feedback provision, peer and self-assessment, and using assessment information to adapt instructional strategies [24]. These practices are guided by

teachers' beliefs about the role of assessment in facilitating student learning, often viewing formative assessment as a dynamic process that seamlessly integrates assessment with instruction to create a collaborative learning environment [13]. Despite the congruence between beliefs and desired practices, challenges remain. Misconceptions about the nature of formative assessment often impede its effective implementation, with some teachers equating it with traditional summative assessments that focus on grades rather than developmental feedback [35].

In addition, how teachers conceptualize and apply formative assessment strategies is often influenced by multiple individual and situational factors, such as teaching experience, educational background, and institutional support [32]. Teachers with more experience or specialized training can typically incorporate formative practices more effectively into their teaching. Conversely, teachers who lack professional development opportunities often struggle to bridge the gap between theory and practice. Institutional factors, such as high-stakes testing and rigorous curriculum requirements, further complicate efforts to prioritize assessment for learning over assessment of learning. Resource constraints, including large class sizes and limited access to technology, exacerbate these challenges, making it difficult for teachers to effectively implement formative assessment [31,35].

Although much research has explored the complex relationship between EFL teachers' beliefs and formative assessment practices, there are still lacunae in understanding how EFL teachers can translate theoretical perceptions into practice in the Chinese context [33]. Additionally, limited research has explored how systemic challenges influence the dynamic interplay between teachers' beliefs and practices.

## 3. Methodology

### 3.1 Participants

This study involved university EFL teachers from various higher education institutions in a province in China. Participants were recruited using a combination of snowball sampling and in-person outreach at academic conferences. The majority of participants were invited through referrals, with the survey link shared via social media by colleagues and their professional networks. In addition, two annual academic conferences related to English language teaching and translation were held between October and November 2024. During these conferences, the researchers approached potential participants in informal settings such as tea breaks, meals, and networking sessions to explain the study and invite participation. Before completing the questionnaire, participants were informed about the study's aims, procedures, voluntary nature, data confidentiality, and their right to withdraw. For in-person invitations, verbal briefing and confirmation were obtained. For electronic invitations, a consent form was shown at the beginning of the online survey, and only those who agreed were allowed to proceed. To ensure data validity, the survey platform was set to prevent duplicate submissions from the same IP address, and all responses were manually screened to exclude incomplete or carelessly filled-out questionnaires.

A total of 322 questionnaires were initially collected, with 34 invalid responses removed, resulting in 288 valid questionnaires. The participants included 27 males (9%) and 261 females (91%). Regarding academic titles, 187 participants were lecturers (65%), 79 were associate professors (27%), and 22 were full professors (8%). In terms of teaching experience, 34 participants (12%) had 1–5 years of experience, 66 (23%) had 6–10 years, 97 (34%) had 11–15 years, and 91 (32%) had over 16 years of experience. Table 1 summarizes the demographic information of the participants.

### 3.2 Instruments

The data were collected using a self-designed questionnaire and semi-structured interviews.

The questionnaire was developed based on the research objectives and consisted of multiple-choice questions. These questions aimed to capture teachers' beliefs about formative assessment, the frequency of formative assessment use, the formative assessment strategies, the use of formative assessment results, and the challenges they encountered. An open-ended question at the end allowed participants to share their experiences and propose improvement plans for future

**Table 1. Demographic information of the participants.**

| Category | Options | Respondent (%) |
|---|---|---|
| Gender | A. Male | 27 (9%) |
| | B. Female | 261 (91%) |
| Academic Title | A. Lecturer or below* | 187 (65%) |
| | B. Associate Professor | 79 (27%) |
| | C. Professor | 22 (8%) |
| Teaching Experience | A. 1–5 years | 34 (12%) |
| | B. 6–10 years | 66 (23%) |
| | C. 11–15 years | 97 (34%) |
| | D. Over 16 years | 91 (32%) |

*Note.* *This category includes positions such as instructors, teaching assistants, and roles equivalent to Assistant Professors in international systems.

formative assessment practices. For ease of distribution, collection, and analysis, the questionnaire was distributed in an electronic form via a popular survey platform in China.

Semi-structured interviews were conducted to better understand teachers' beliefs and practices about formative assessment and the implementation challenges. The interview guidelines comprised the following questions:

1. Teachers' understanding of formative assessment and its role.

2. How formative assessment is implemented in their classrooms.

3. The use and feedback of formative assessment results.

4. Challenges or difficulties in implementing formative assessment.

Each interview lasted approximately 40 minutes and was recorded for analysis.

### 3.3 Procedure

The survey was distributed electronically through social media groups over a period of 60 days during the end of Year 2024. A total of 322 responses were collected, of which 288 were considered valid after excluding incomplete and casually filled-out surveys.

Simultaneously, semi-structured interviews were conducted with 11 teachers who had provided their contact information in the questionnaire and expressed willingness to participate. Among them, two were professors, four associate professors, and five lecturers or below. The first researcher conducted the interviews face-to-face or via video call. All interviews were recorded to ensure accuracy and allow for subsequent transcription and analysis.

### 3.4 Data analysis

Quantitative data from the questionnaire were analyzed using descriptive statistics to summarize participants' demographic information and trends in formative assessment beliefs and practices. For the open-ended question, qualitative content analysis was used to extract themes related to teachers' experiences and suggestions.

The qualitative data from the interviews were transcribed and thematically analyzed. The analysis involved coding the transcripts to identify patterns and recurring themes related to teachers' beliefs, practices, and challenges in implementing formative assessment. This mixed-methods approach provided a comprehensive understanding of the research questions, combining numerical trends with detailed contextual insights.

## 3.5 Ethical considerations

This study did not involve any medical, physiological, or psychological interventions and was classified as minimal-risk educational research. According to the academic research guidelines at the researchers' institution, formal ethics approval was not required for such studies. A formal statement confirming this exemption was obtained from the university's Academic Committee.

Informed consent was obtained from all participants prior to data collection. For the online questionnaire, participants were presented with a written informed consent form at the beginning of the survey. This form outlined the purpose of the study, research procedures, voluntary participation, data confidentiality, and the right to withdraw at any time. Only those who confirmed their consent by checking the agreement box were able to proceed with the questionnaire.

For the interviews, verbal informed consent was obtained prior to the start of each session. The researcher explained the study's purpose, the voluntary nature of participation, confidentiality safeguards, and the participants' right to withdraw at any time. This verbal consent was audio-recorded at the beginning of each interview to serve as documentation.

All data were collected anonymously. No personally identifiable information was recorded or reported, and all responses were analyzed in aggregate form to ensure participant confidentiality.

# 4. Findings

## 4.1 Teachers' beliefs of formative assessment

**4.1.1 Is formative assessment important in EFL teaching?** Teachers' attitudes toward formative assessment reflect their recognition of its importance in EFL teaching. The results reveal that 72% of teachers consider formative assessment "very important," 25% view it as "important," and only 4% regard it as "Unsure." Notably, no teacher rated it as unimportant or completely unimportant, indicating that almost all teachers highly value its role in university English education (Table 2, Table 3, Table 4).

Cross-analysis based on various teacher characteristics reveals some interesting patterns. Teachers with 1–5 years and 6–10 years of experience predominantly (82%) find formative assessment "very important". This suggests that

**Table 2. Teachers' attitudes towards FA in terms of teaching experience.**

| Option | 1-5 Years | 6-10 Years | 11-15 Years | ≥16 Years |
|---|---|---|---|---|
| Very Important | 28 (82%) | 54(82%) | 64(66%) | 60(66%) |
| Important | 4(11%) | 10(15%) | 27(28%) | 30(33%) |
| Unsure | 2(6%) | 2(3%) | 6(6%) | 1(1%) |

**Table 3. Teachers' attitudes towards FA in terms of academic title.**

| Option | Lecturer and Below | Assoc. Prof. | Full Prof. |
|---|---|---|---|
| Very Important | 146(78%) | 43(54%) | 17(77%) |
| Important | 35(18%) | 31(39%) | 5(23%) |
| Unsure | 6(4%) | 57%) | / |

**Table 4. Teachers' attitudes towards FA in terms of gender.**

| Option | Male | Female | Total |
|---|---|---|---|
| Very Important | 17(63%) | 189(72%) | 206 (72%) |
| Important | 8(30%) | 63(24%) | 71 (25%) |
| Unsure | 2(7%) | 9(3%) | 11 (4%) |

early-career teachers, likely to have recently undergone training in modern educational theories, are more inclined to embrace this approach. For mid-career teachers (11–15 years), 66% consider it "very important," while 28% find it "important," showing a generally positive but slightly less enthusiastic attitude compared to their younger colleagues. Experienced teachers (≥16 years) also exhibit strong support, with 66% rating it as "very important" and 33% as "important". Overall, despite some variation, there is a clear consensus among teachers across different career stages regarding the significance of formative assessment.

Different academic titles indicate varying levels of recognition regarding the importance of formative assessment. Among lecturers, 78% perceive formative assessment as "very important," with 18% finding it "important" and 4% considering it "unsure." Lecturers, who often focus on integrating theory with practice, demonstrate a high level of endorsement. Associate professors show slightly lower but still substantial support, with 54% viewing it as "very important" and 39% as "important." Despite having a smaller sample size, professors exhibit strong agreement, with 77% rating it as "very important" and 23% as "important." These differences may stem from varying priorities and experiences across academic titles, with lecturers being more open to modern educational practices and senior academics relying on extensive teaching experience.

Gender also plays a role in shaping perceptions of the importance of formative assessment. Female teachers generally show a higher endorsement rate, with 72% considering it "very important" compared to 63% of male teachers. Additionally, 24% of female and 30% of male teachers find it "important," while only 3% of female and 7% of male teachers rate it as "average." Although the sample size for male teachers is relatively small, this data suggests that female teachers tend to place slightly greater importance on formative assessment. Nonetheless, both genders predominantly hold positive views on its significance in enhancing teaching and learning outcomes.

The findings reveal a widespread acknowledgment of the importance of formative assessment among university English teachers. Early-career and female teachers exhibit slightly higher endorsement rates, possibly reflecting recent training and a greater inclination towards modern educational practices. These insights provide valuable guidance for promoting and implementing effective formative assessment strategies in university English education.

**4.1.2 What is formative assessment?** While the findings reveal a strong belief among teachers in the importance of formative assessment, the teachers differ in their conceptualization of it. The results showed that most respondents agreed that formative assessment involves facilitating student learning in various ways (Table 5). Specifically, 82% (n = 235) of the respondents believed that formative assessment involves periodical quizzes or assignments to understand student learning and improve pedagogical approaches, representing the highest percentage of consensus. 74% (n = 214) of the respondents noted that formative assessment involves using informal methods to monitor learning progress and guide pedagogical adjustments. Another 72% (n = 206) of respondents emphasized the importance of student self-assessment and peer assessment as contributing to enhancing students' reflective skills and learning outcomes. Meanwhile, 66% (n = 189) of the respondents agreed that formative assessment is a theory that promotes student learning through continuous feedback. It is worth noting that despite a more comprehensive overall understanding, 50% of the respondents (n = 143) indicated that they were unsure or unclear about the concept of formative assessment, suggesting that cognitive ambiguity exists among teachers as to what constitutes formative assessment.

**Table 5. Teachers' understanding of the concept of FA.**

| Option | Respondent | Percentage |
|---|---|---|
| It is a theory that promotes student learning through continuous feedback. | 189 | 66% |
| It involves informal methods to monitor learning progress and guide instructional adjustments | 214 | 74% |
| It consists of periodical quizzes or assignments to understand students' learning status and improve teaching. | 235 | 82% |
| It involves student self-assessment and peer assessment to enhance self-reflection skills and learning outcomes. | 206 | 72% |
| Uncertain/Not clear | 143 | 50% |

This diversity of understanding of formative assessment was further exemplified in the subsequent teacher interviews, which presented theoretical and practical approaches. A few teachers regarded formative assessment as an assessment theory composed of complicated mechanisms. For example, T5 stated:

*Formative assessment is an assessment theory that guides our education assessment. It involves understanding how continuous feedback can shape student learning over time. (T5)*

In contrast, other teachers emphasized a more practical approach

*Formative assessment is the collection of all assessment activities in and out of our class every day. It's the small, actionable steps we take to check in with students and adjust our teaching accordingly. (T11)*

*I see formative assessment as the activities in that teachers collect and evaluate evidence of students' learning both in and out of class, and then provide feedback to the students. (T6)*

These contrasting understandings of formative assessment suggest that many teachers lack a deep grasp of its nature, which might lead to great variety in practices.

**4.1.3 What is the focus of formative assessment?** The survey results showed diverse views when exploring the views of college English teachers on what is measured by formative assessment (Table 6). Specifically, 82% of the respondents (n = 235) believed that engagement and collaboration skills were important aspects of formative assessment, and 74% of the respondents (n = 212) emphasized the importance of learning attitudes and effort. In comparison, 69% (n = 198) of the teachers pointed out that language skills were likewise one of the key components of assessment. Although mastery of knowledge points was also agreed upon by 58% of the respondents (n = 167), this was lower than the other options, showing that teachers placed more emphasis on developing general skills and attitudes. Another 29% of respondents (n = 83) chose the "other" option, suggesting that there may be more individualized considerations. Overall, college English teachers view formative assessment as a comprehensive assessment tool that focuses not only on students' language skills but also on their engagement, collaboration, and learning attitudes, reflecting the educational philosophy of promoting students' all-round development. However, this diversity might also reflect teachers' uncertainty about the focus of formative assessment.

This diversity of formative assessment focus was also fully realized in the in-depth interviews with the teachers. Though all agree the students should be assessed comprehensively, some believe language skills should be the assessment focus and be evaluated frequently to help students improve their learning, others suggest an emphasis on the students' attitudes to improve their learning habits and behaviors.

*Formative assessment in class can be divided into several parts. But the most important part is to evaluate their [the students'] improvement of language proficiency during the whole semester. (T11)*

**Table 6. Teachers' beliefs on the FA focus.**

| Option | Respondent | Percentage |
|---|---|---|
| Language skills (listening, speaking, reading, writing) | 198 | 69% |
| Participation and collaboration abilities | 235 | 82% |
| Learning attitude and effort level | 212 | 74% |
| Mastery of knowledge points | 167 | 58% |
| Others | 83 | 29% |

T11 represents quite a few of the interviewees. To T11, formative assessment in class was to make the process of student learning tangible and observable, including their engagement, teamwork, exercises, and the language learning results. But the learning results is most important to her. So her assessment focus was on this part, and gave many quizzes in the formative assessment process.

While other teachers differ in their understanding of the focus in formative assessment:

*I believe, besides checking students' learning progress once or twice, formative assessment should really focus on monitoring things like [the students'] emotions and attitudes toward learning, not just how much knowledge they've picked up at the moment. (T8)*

To T8, the formative assessment was to help the students stay persistent in their learning. Therefore, she placed more emphasis on evaluating the frequency of task completion, the amount and regularity of out-of-class study time, and whether students adhere to their study plans. T8's perspective is not an isolated case; a few teachers hold the same opinion.

**4.1.4 What is formative assessment used for?** The survey results indicate that a majority of teachers hold positive beliefs regarding the impact of formative assessment on students' learning behaviors, performance, and affection (Table 7).

As shown in Table 7, teachers believe that the most significant purpose of formative assessment is to promote class participation (92%). This is similar to Cole's (2012) study. This indicates that incorporating classroom performance into regular grades and linking it with course outcomes can substantially increase student engagement. As a participant noted,

*As soon as students know that their answers or participation in an activity can earn points which count toward their final grade, they immediately become active and interested.(T4)*

In addition, "enhancing learning motivation" (85%) and "improving learning habits" (88%) are also important manifestations of the functions of formative assessment. These findings suggest that formative assessment strengthens students' intrinsic drive, which is beneficial for altering student learning outcomes.

However, it also shows that formative assessment is not as highly recognized in areas such as "boosting student confidence" (72%), "enhancing teamwork" (77%), and "improving teacher-student relationships"(64%). T8 reflects, "I conduct formative assessment mainly to understand and monitor students' learning progress". She adds that subjective aspects like emotions are not easy to capture and are not something she usually pays attention to. This suggests that teachers' perceived effects of formative assessment on students' emotional attitudes may not be as strong as its impact on students' learning behaviors and academic performance.

**Table 7. Teachers' beliefs in the purpose of formative assessment.**

| Option | Respondent | Percentage |
|---|---|---|
| Improving classroom engagement | 265 | 92% |
| Enhancing learning motivation | 246 | 85% |
| Improving teacher-student relationships | 184 | 64% |
| Boosting students' confidence | 207 | 72% |
| Improving learning habits | 254 | 88% |
| Enhancing teamwork | 223 | 77% |
| Others | 87 | 30% |

## 4.2 Teachers' practices of formative assessment

### 4.2.1 Frequency of formative assessment implementation.

The frequency of formative assessment implementation reflects teachers' efforts to consistently monitor student learning progress, adjust teaching strategies, and align instructional practices with course objectives. According to the survey, 57% of teachers determine formative assessment implementation frequency based on their own teaching experience, followed by adherence to institutional policies (41%) and alignment with course objectives (32%). This reliance on subjective personal experience highlights a lack of standardized practices, which may result in inconsistencies in how learning progress is tracked across teachers.

Survey data also reveals substantial variability in teachers' actual implementation frequency. While 52% of teachers conduct formative assessment 6–10 times per semester, 32% implement it fewer than 5 times, and 8% more than 20 times. Further analysis shows that teaching experience notably influences the frequency. Teachers with 1–5 or 6–10 years of experience predominantly conduct fewer than five assessments per semester, whereas those with over 11–15 or above 16 years of experience more frequently implement formative assessment, with some conducting it more than 20 times. This trend suggests that experienced teachers, having developed effective pedagogical strategies over time, are more confident and adept at integrating formative assessment into their instructional routines. The interview with the teachers verified this view. T 3, a 4-year-experience teacher, said:

*Actually, the number of formative assessments [I conducted] is more than 6 times shown in the final report. I mean, there are many different formative assessment evidences. However, if I had to evaluate and keep a record every class or every week, I'd never complete that. So, my formative assessment is kind of random—also with the help of the teaching platforms, otherwise I cannot complete [the formative] assessment for all students. (T3)*

It reflects that while novice teachers have acquired a certain level of theoretical knowledge regarding formative assessment, they may not yet be proficient in the operational activities. For example, they may lack a systematic approach to the assessment process (although they have a formative assessment plan), and be uncertain about how to integrate formative assessments into daily instruction, but it also reflects the less experienced teachers are quite used to the technology, that is, the teaching platforms.

On the contrary, experienced teachers are skilled in organizing the assessment and managing the assessment data. For instance, T5, an associate professor, said:

*I think that for formative assessment to really work, it must allow the [assessment] data to guide your teaching and the students' learning, then it is … an effective evaluation. So, I make it a point to evaluate in each class and share data with the students every week—you know, their performance, how well they cooperate, their homework, their extracurricular learning, all of it. I would implement [FA] more than 16 times a semester.(T5)*

To T5, she could conduct formative assessment regularly in class and manage the data properly. This might be explained by that the teachers with higher titles are more skillful in organizing formative assessment activities and integrating it into their teaching.

### 4.2.2 Strategies and technology for formative assessment implementation.

Formative assessment strategies serve as essential tools for teachers to assess student learning progress from multiple dimensions, identify individual learning needs, and provide timely feedback for improvement. Survey results indicate that the most commonly used tools are classroom questioning (100%), individual assignments (96%), group projects (84%), and periodic testing (84%). However, self-assessment (28%) and peer assessment (24%) are utilized less frequently. This might indicate that teachers value both timely communication with students in the classroom and real-time monitoring of student learning, as well as independence and collaboration in student learning. However, the limited use of self- and peer-assessment

strategies possibly reflects that teachers overlook the cultivation of students' self-reflection and self-monitoring skills, as well as their concerns about the effectiveness of peer assessment.

The interview found that, in selecting assessment strategies, teachers are primarily mainly influenced by factors including institute policies, the nature of the course, fellow course instructors, and their personal experiences.

*The institute is implementing formative assessment and asked us to fill out the formative assessment plans before we start teaching the course. They want us to pick more than three evaluation strategies from the list. After discussing with the other teachers who teach the same course, we'll just tick a few options that we agree on.(T7)*

Actually, most universities have developed their own rules for implementing formative assessment, which include evaluation strategies, feedback mechanisms, implementation and supervision. Therefore, teachers could select from the formative assessment strategies prescribed by the university and have the flexibility to adjust the proportions of each assessment strategy in the whole formative assessment. However, in deciding which strategies should be used in the teaching, the course nature, fellow course instructors' opinions, or their personal experiences exert great influence. For example, for teachers like T1, their understanding on the nature of language learning influenced their choice of formative assessment strategies.

*I think language learning is mainly about practice, so I place a lot of importance on evaluation methods that involve oral skills, like in-class exercises and speech presentations. (T1)*

Evidently, the selection of formative assessment strategies is subject to the influence of a variety of factors, including institute rules, the course objectives, fellow instructor opinions, and personal cognition and experiences, requiring a comprehensive and integrated consideration.

Technology also plays a significant role in supporting formative assessment implementation. To encourage universities to adopt their textbooks, publishers have developed teaching platforms for their textbooks, making it easier for instructors to teach. According to the survey, textbook-publisher-supported teaching platforms (68%) and commercial teaching platforms (64%) are the most widely adopted in their formative assessment practices, followed by language learning apps or mini-programs (44%) and social media tools like QQ or WeChat groups (32%). Traditional paper-based materials are rarely used (4%), reflecting a strong preference for digital tools in modern educational contexts.

**4.2.3 Utilization of formative assessment results.** The effective utilization of formative assessment results is central to improving teaching strategies and fostering student development. According to the survey results (Table 8), teachers primarily use formative assessment results to refine their teaching methods (58%), calculate final grades (52%), and motivate classroom participation (44%). However, only a few teachers use formative assessment results to help students understand their learning progress (12%) or promote teamwork (20%). These findings highlight a gap between formative assessment's intended purpose and its current practice application.

**Table 8. EFL teachers' utilization of FA results.**

| Options | Respondent | Percentage |
|---|---|---|
| To refine teaching methods | 167 | 58% |
| To calculate students' course average grades | 150 | 52% |
| To promote classroom participation | 127 | 44% |
| To help students understand their learning progress | 35 | 12% |
| To meet the institute's requirements | 109 | 38% |
| To encourage student teamwork | 57 | 20% |
| Other | 23 | 8% |

Notably, 38% of teachers report that formative assessment results are primarily used to meet the institutional requirements rather than to enhance pedagogical or learning outcomes. While this approach provides a tangible output for institutional accountability, it diverges from the intended purpose of formative assessment to foster student-centered learning and self-regulation. The findings suggest that the current utilization of formative assessment results only partially fulfills its intended function, with significant room for improvement in aligning its use with pedagogical goals.

In the interview, teachers stated they know the purpose of formative assessment is to enhance teaching and learning. However, in reality, their use of formative assessment results has somewhat deviated from the intended direction. T4 said:

*The outcomes of formative assessment must accurately reflect students' actual learning conditions on an ongoing basis. They should distinguish between students who are genuinely engaged in their studies and those who are less engaged. By providing high scores or low scores, I want to encourage or punish [the students] so as to push them to become more involved in their learning. (T4)*

T4's statement reflects that some teachers use the results of formative assessments as a tool to reflect students' daily learning performance and attitudes and differentiate students based on their level of engagement in learning through scores.

*To be frank, in theory, each set of data should be individually fed back to the students. However, in practice, I haven't carried out this step yet currently. Due to the large number of students, these data are mostly just kept on file to meet the institute's inspection requirements. (T2)*

For teachers like T2, in practice, due to the large number of students, they find it difficult to provide personalized feedback for each assessment. As a result, these formative assessments often end up being more about meeting the school's inspection requirements rather than fully utilizing its potential to motivate and guide student learning.

### 4.3 Challenges in implementing formative assessment

The questionnaire and the interviews with university English teachers revealed several key challenges they face in implementing formative assessment despite its recognized importance in enhancing student learning and aligning with national educational reforms. Formative assessment in university English instruction was mentioned as early as 2003 in the *National College English Teaching Guideline* (Guo & Yang, 2003). In 2018, the State Council further emphasized the need to "reform educational evaluations" to promote the fundamental task of cultivating talents. However, despite these guidelines, the practical application of formative assessment in university English teaching remains far from ideal.

Table 9 shows that "heavy assessment workload" (76%) is the most common challenge teachers face, significantly higher than other options. This indicates that frequent assessment tasks in large-class teaching environments impose a substantial burden on teachers.

"Lack of objectivity and fairness in self- and peer-assessment" (60%) ranks second, indicating teachers' concerns about the reliability and credibility of these assessment forms. Following closely is "obscure assessment criteria" (52%), which reflects teachers' confusion about how to establish and apply clear and consistent assessment standards. "Inadequate technical support" (44%) is also a significant barrier, suggesting that existing technical support does not fully meet teachers' needs, potentially limiting the effective implementation of formative assessments. While "insufficient time" (32%) and "low student engagement" (24%) have relatively lower percentages, they remain significant factors. These results highlight the challenges teachers face in balancing high-quality teaching and assessment within limited time frames, as well as the issue of some students not actively participating in formative assessment activities. Similarly, "lack of training" (24%) indicates that the professional development opportunities provided by institutes may be insufficient to equip teachers with the necessary skills and knowledge to effectively conduct formative assessments.

**Table 9. Challenges in FA implementation.**

| Options | Respondent | Percentage |
|---|---|---|
| Insufficient time | 92 | 32% |
| Inadequate technical support | 127 | 44% |
| Low student engagement | 69 | 24% |
| Obscure assessment criteria | 150 | 52% |
| Heavy assessment workload | 219 | 76% |
| Lack of training | 69 | 24% |
| Lack of objectivity and fairness in self- and peer-assessment | 173 | 60% |
| Others | 12 | 4% |

The interview results are strongly corroborate the survey results, reaffirming key challenges including large class sizes, lack of technical support, excessive arbitrariness in student evaluations, and unclear assessment criteria.

Large class sizes were identified by most respondents as a reason for the challenges in implementing formative assessment. Generally, the number of students in a university English class varies from 30 to over 100, so even a simple assessment activity is quite time-consuming for teachers. They frequently cited a large number of classes as a major obstacle, making it greatly difficult to effectively carry out formative assessments, particularly when it comes to providing individualized feedback to such a number of students they are responsible for. T1 stated:

*I've taught four large classes [this semester], with a total of over 350 students. I would assign their homework, such as translations and essays, through the teaching platform. If I were to grade them all manually, it would be impossible to finish even without taking any breaks. (T1)*

From the responses of T1, it can be seen that excessively large class sizes lead to an overwhelming workload for teachers in conducting formative assessments. This makes it difficult to provide timely and personalized feedback to a large number of students, thereby affecting the effective implementation of formative assessment.

Another significant challenge is the lack of sufficient technological support. While digital platforms and educational technologies are increasingly integrated into teaching, teachers reported that available tools and platforms often fail to meet their assessment needs.

*The teaching platforms are really effective when used for objective questions, but for subjectively evaluated questions, I only use their results as a reference. For translation exercises, there are often many misjudgments, especially for more creative students, as machine grading is hard to achieve accuracy. (T7)*

*You know, many students' work might be… supported by AI. But [the platform] could not determine whether the work submitted by students is genuinely their own. (T10)*

This reflects that teachers lack sufficient technical support when implementing formative assessments, especially in ensuring the authenticity and reliability of the data.

Additionally, teachers face difficulties in setting clear and scientific assessment criteria. Although most teachers clearly outline the criteria for formative assessment at the beginning of the course, interviews revealed that teachers feel the standards are somewhat vague in practice. The following comments from two teachers are representative of this sentiment:

*Classroom performance is the area I find most confusing and challenging… I just know that it involves aspects like student interaction, participation, and task completion. But my assessment tends to be quite general. I haven't figured*

*out how to do it well, nor do I know exactly where to start. I also haven't seen any particularly convincing practices [on this aspect] from others. (T6)*

*The institute requires that our formative assessment include at least three strategies like scored assignments, group work, etc. How do we evaluate the performance in group work? How can we break it down into assessment dimensions that are easy for teachers to operate and perceived by students as fair, just, and transparent? No one has told me. (T8)*

Teachers feel that the evaluation is "rather general" and lacks scientific "dimensions," indicating that some assessment criteria are relatively vague and difficult to apply precisely. This lack of clear criteria can lead to inconsistencies in assessment practices and reduce the overall effectiveness of formative assessment in promoting learning improvement.

Furthermore, challenges related to self-assessment and peer assessment were also prominent in the interviews. Teachers noted that students often struggle to conduct objective self-assessments. Personal biases, such as emotional reactions or self-perception distortions, frequently influence these assessments. In peer assessments, social factors such as friendships or competitive dynamics among students can lead to biased evaluations, which are "*very annoying and upset their classmates*"(T7). These biases undermine the objectivity and fairness of peer evaluations, which, in turn, reduces the reliability of feedback intended to guide student improvement.

Moreover, the students lack the training to perform self- and peer-assessments effectively. Many students are unfamiliar with the standards and criteria for evaluation, "*particularly when it comes to assessing complex skills such as critical thinking or creative abilities*" (T10). Teachers believe students' indifferent attitudes and limited evaluation skills, along with the influence of personal relationships, significantly impact the objectivity and fairness of the peer assessment process.

These findings suggest that several practical barriers prevent its full realization. Teachers' heavy workloads, insufficient technical resources, unclear evaluation standards, and challenges with student self- and peer-assessment all contribute to the difficulties in implementing formative assessment effectively.

## 5. Discussion

### 5.1 Teachers' beliefs of formative assessment

This study reveals the Chinese university English teachers' beliefs about formative assessment in a few aspects. A strong majority recognized the importance of formative assessment. This may be attributed to the Ministry of Education's explicit requirements for formative assessment [36] and the popular claim that formative assessment is the core mechanism of "assessment as learning" [37]. However, while previous studies often emphasize the role of formative assessment in fostering critical thinking [13,38], the teachers in the present study focused more on its foundational function of facilitating classroom engagement or participation. This discrepancy may reflect differences in the positioning of assessment goals in different educational systems.

Regarding conceptual understanding, a significant gap was evident. The widespread simplification of FA as synonymous with "periodical tests" (82%) starkly contrasts with established FA theory, which emphasizes its ongoing, process-oriented, and multi-method nature [1,39]. This simplistic view, also documented among Chinese secondary EFL teachers [40], perpetuates the cognitive inertia of traditional testing cultures [41] and indicates a substantial deficit in the "knowledge and understanding" dimension of assessment literacy [42]. Meanwhile, quite a few (74%) of teachers understood that participants interpreted it as informal classroom observation, a practice-oriented cognitive model. This conceptual understanding might lead the teacher to rely mostly on immediate, intuitive feedback methods in their teaching practices while overlooking systematic and planned assessment approaches. As noted by Dolin et al. [43], teachers who lacked a unified cognitive framework were more likely to fall into the operational dilemma of "evaluation for evaluation's sake"; This resonates with findings that Chinese EFL teachers often lack theoretical training and thus rely heavily on spontaneous judgment [12], which undermines planned, criterion-referenced feedback, leading to the operational dilemma of

"evaluation for evaluation's sake." [43]. Within the formative assessment literacy (FAL) framework [44], the lack of conceptual clarity may impair the teacher's ability to purposefully align assessment with learning goals and activate students as learning agents.

Furthermore, teachers in this study demonstrated inconsistent understandings of FA focus. While a large proportion highlighted language proficiency and classroom engagement, few teachers focused on the assessment of affective and metacognitive dimensions. This ambiguity has led to notable divergence in assessment tool choices, with some practitioners favouring quantitative rating scales while others rely on qualitative observations or interpretive rubrics. Such inconsistency resonates with the identification by Heritage [39] and Bennett [45] identification of defining clear learning progressions and success criteria as a persistent hurdle in formative assessment implementation. Addressing this requires collaborative efforts to co-construct contextually appropriate formative assessment goals, clarify indicators, and develop mixed-method tools to enhance reliability and interpretability of results [46].

Perhaps the most striking finding concerning beliefs relates to the perceived purpose of FA. While most participants valued FA for supporting student learning, few perceived it as a diagnostic tool for instructional improvement. This reflects a unidimensional understanding that stands in contrast to the dual-purpose model commonly advocated in the literature, wherein formative assessment informs both teaching and learning [45,47,48]. Such asymmetry may be attributed to insufficient data literacy among teachers, as well as systematic disconnections between assessment outcomes and curricular decisions. Without institutional conditions that support reflective teaching informed by assessment evidence, the potential of FA for instructional refinement remains untapped [42]. Therefore, professional development efforts should not only enhance conceptual understanding but also focus on developing teachers' capacity to use FA results for adaptive teaching. To address this issue, policymakers should prioritize incorporating comprehensive formative assessment training into pre-service and in-service teacher education programs.

## 5.2 Teachers' practices of formative assessment

The study also reveals the characteristics of Chinese English teachers in their formative assessment practices. The results indicated that the implementation frequency correlates positively with teaching experience. This is consistent with [49] findings that the accumulation of teaching experience contributes to the routinization of assessment. However, teachers with large class sizes generally reported difficulty in sustaining high-frequency assessments, which validates the hypothesis that there is a feasibility tipping point for formative assessment [43]. This challenge appears amplified compared to settings frequently documented in Western FA literature [13], likely due to the scale of Chinese higher education. Our findings thus directly address a gap concerning the practical realities of enacting FAL in large EFL classes, advocating for context-specific solutions like teaching assistants, adjusted class sizes, or the "student feedback partner" model to alleviate the burden and refocus on learning.

In terms of tools and strategies, most teachers relied on conventional teacher-centered methods such as questioning and individual assignments. Practices requiring higher student autonomy such as self-assessment and peer review were less frequently employed, echoing findings that many EFL teachers remain skeptical of these approaches' validity [50]. This hesitancy starkly contrasts with substantial evidence demonstrating their efficacy when properly structured, scaffolded, and supported by clear criteria [18,51]. Teachers often lack confidence in enabling peer feedback, in part due to vague criteria and the absence of scaffolded instruction [47]. Additionally, the reported lack of clarity in evaluation criteria further exacerbates this issue, hindering the translation of any assessment results into concrete improvement steps. This highlights a specific deficiency in teachers' FAL concerning empowering learners as assessors. Addressing these problems require comprehensive support focusing not only on self assessment or peer assessment techniques but, crucially, on co-developing clear criteria, providing calibration exercises, and integrating digital tools where appropriate. Given the prevalence of large classes and exam pressures in Chinese universities, institutional efforts are needed to embed peer feedback into curriculum design and to offer teachers localized training and assessment templates tailored to the Chinese EFL context.

The paradoxical phenomenon of technology application highlights the complexity of implementing formative assessment in the digital era. Although most teachers use digital platforms, they generally encounter technical barriers, resulting in the dilemma of "high utilization - low effectiveness." This confirms findings that when the complexity of tools exceeds teachers' technological capabilities, assessment practices tend to degenerate into simple data collection [52,53]. Crucially, it reveals a gap in the "design/implementation" dimension of feedback literacy: the ability to strategically select and leverage technology not merely for data collection, but for generating timely, actionable feedback, facilitating interactive learning dialogues, and enabling teachers to track progress efficiently [47]. Our findings suggest that simply providing platforms is insufficient; institutional support must explicitly focus on building teachers' assessment literacy in the digital realm, including technical training, pedagogical guidance on using data dashboards, and access to tools designed for rich formative interactions rather than just summative recording.

Despite widespread agreement on FA's role in facilitating learning [54], the predominant uses were for grading and teaching adjustments, with minimal application for self-regulated learning (12%). This contradicts the argument that "the purpose of FA is to identify learning goals and provide feedback rather than compare students using a normative approach"[30, p. 2]. The limited application of formative assessment results for fostering self-regulated learning or cognitive development reveals a deficit in key FAL, that is, the capacity to generate feedback that moves learning forward cognitively [55] and the systematic use of evidence for substantive instructional adaptation were absent [47]. This may be associated with institutional pressures: teachers tend to administer formative assessments to fulfill institutional requirements, resulting in data being mired in archival status (as in T2's case).The results suggest that FA in practice remains entangled with summative traditions, a tension exacerbated by institutional accountability pressures. Embedding formative feedback quality into teacher appraisal criteria may reinforce more pedagogically aligned practices.

At the same time, this study's reliance on self-reported data may not fully reflect actual classroom practices. Previous research has shown that self-reports often overlook the actual forms and frequency of assessment, and that discrepancies exist between teachers' perceived skills and their reported use of assessment tools [56]. These inconsistencies suggest that social desirability or self-perception bias may affect data accuracy, underscoring the need for future studies to triangulate findings with observations or artifacts.

## 5.3 Challenges in implementing formative assessment

The research revealed several challenges in implementing formative assessment, most notably the overwhelming workload and time constraints. This result aligns with research that teachers' dual roles as pedagogical guides and administrative assessors lead to cognitive overload [37]. Similar findings were reported in different EFL contexts, where time constraints and heavy teaching loads led EFL teachers to prioritize efficiency over depth in formative assessment implementation [57].While existing research confirms large class sizes as barriers, the findings of this study further suggest this challenge is amplified because teachers have to juggle two conflicting goals when doing assessments: to help students improve and to meet institutional reporting requirements. To address this issue, universities can provide additional teaching assistants, consider smaller class sizes, or adopt the "student feedback partner" model, where trained students assist with initial data collection, allowing teachers to focus more on individual student progress and offer personalized feedback.

Another challenge identified in the study was the insufficient technical support for formative assessment. While digital platforms are increasingly used, teachers reported that the available tools often did not meet their needs for detailed, interactive formative assessments. This is consistent with findings that in digital learning environments, teachers frequently encounter difficulties in aligning available assessment technologies with pedagogical intentions [58]. Moreover, effective technology use in formative assessment is contingent not only on the availability of tools but also on teachers' confidence, training, and institutional support [59]. These studies collectively suggest that the mere presence of technology is insufficient without a supportive infrastructure.

The lack of transparent and standardized evaluation criteria was also a significant challenge. Teachers found it difficult to establish reliable assessment standards for those abstract aspects of evaluation, such as learning attitudes and classroom performance. This inconsistency undermines the effectiveness of formative assessment and makes it harder to provide accurate feedback. Similar challenges have been observed in various EFL contexts, where teachers reported struggling to develop consistent criteria for non-academic dimensions, leading to subjective evaluations and reduced assessment reliability [3,60]. To address this, teachers could form teams to collaboratively discuss and share assessment experiences, focusing on evaluating abstract aspects, thereby enhancing the effectiveness and consistency of assessments. Additionally, teachers could fully utilize qualitative assessment methods, such as observation notes, student interviews, and portfolio reviews, to gain deeper insights into students' learning processes and achievements [61,62]. This approach may help teachers comprehensively understand students' growth and development, enabling them to develop more personalized and effective teaching strategies. However, while the findings offer valuable insights into EFL teachers' formative assessment practices, the geographical focus on a single province may limit the generalizability of the results across diverse Chinese contexts.

Finally, the challenges with self-assessment and peer assessment indicate that students often struggle to evaluate their own and their peers' performance objectively. These findings align with previous studies showing that EFL teachers tend to underutilize self- and peer assessment practices in formative assessment [63–65]. The underuse of self- and peer assessment may be due to students lacking the necessary skills to provide accurate assessments, resulting in inconsistencies and potential biases [66,67]. Additionally, cultural factors significantly influence peer assessment, as students from different cultural backgrounds may differ in their acceptance of critique and assessment styles, potentially limiting the effectiveness and scalability of peer assessment strategies [68,69].

To clarify the interconnections among the key findings, we present a visual model that illustrates how formative assessment unfolds in the Chinese EFL context (Fig 1). The model integrates teachers' beliefs, reported practices, challenges, and conditions that support effective implementation, and it also reflects their dynamic interplay shaped by contextual factors.

As presented in Fig 1, this model shows how teachers' beliefs shape their their practices, but this process is constrained by contextual challenges such as heavy workload and time constraints. Enabling conditions, such as professional

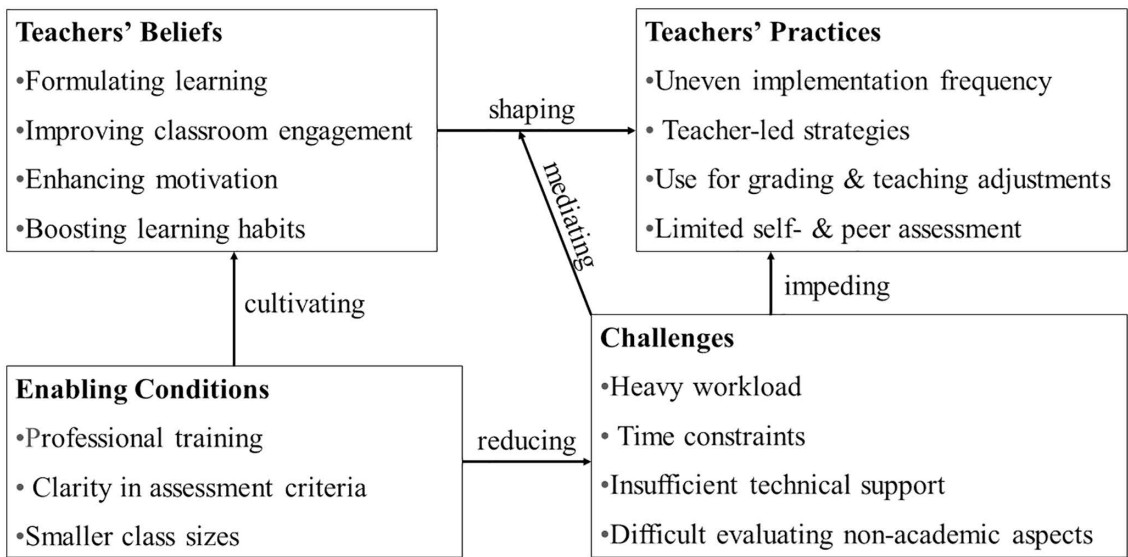

**Fig 1. A conceptual model of FA implementation in Chinese EFL contexts.**

training and smaller class sizes, help mitigate challenges and reinforce or cultivate teachers' beliefs, forming a dynamic system that affects how formative assessment is enacted.

Based on the proposed conceptual model, several policy implications emerge for enhancing formative assessment practices among Chinese EFL teachers. To begin with, targeted professional development programs are needed to strengthen teachers' understanding about formative assessment and improve their capacity to address practical challenges. Second, institutional reforms are needed to relieve contextual constraints, such as large class sizes and heavy teaching loads, by offering organizational support and greater flexibility in assessment design. Finally, more explicit and operationalized assessment criteria, especially for non-academic domains, should be developed to guide teachers in conducting more valid and reliable formative assessment. These efforts can jointly support a more coherent and sustainable FA culture in Chinese universities.

## 6. Limitations

This study has several limitations. First, the data were collected from university EFL teachers in a single province in China. Although participants represented a range of institution types and professional backgrounds, the findings may not fully reflect formative assessment beliefs and practices across other regions. Regional differences in educational policy, institutional support, and assessment culture may shape how formative assessment is perceived and enacted. Thus, the generalizability of the results is limited.

Another limitation lies in the study's reliance on self-reported data through questionnaires and interviews. Such methods may be prone to response bias, as participants could offer socially desirable answers or overstate their engagement with formative assessment, particularly under the influence of current national policy discourse. Without classroom observations or documentary evidence (such as student work or feedback artifacts), it is difficult to verify whether reported practices align with actual behaviors.

Future research should consider multi-regional sampling and incorporate observational and artifact-based data to gain a deeper understanding of formative assessment implementation and reduce potential self-reporting bias.

## 7. Conclusion

This study investigated university English teachers' beliefs, practices, and challenges in implementing formative assessment. The findings reveal that while the majority of teachers hold positive beliefs regarding the importance of formative assessment, implementation practice remains uneven. Teacher-led assessments are far more common than self- or peer assessment, and the formative assessment results are often used for grading or teaching improvement rather than promoting self-regulated learning. Challenges such as time constraints, workload, and insufficient technological support further hinder consistent implementation.

By uncovering the belief-practice gap and practical barriers in the Chinese context, this study contributes to the growing body of research on formative assessment in EFL settings. It underscores the importance of targeted professional development, collaborative teacher inquiry, and context-sensitive strategies to enhance formative assessment practices, especially in exam-driven educational cultures. Efforts should also be made at the policy level to reduce summative dominance and foster a more supportive environment for formative assessment integration.

## Supporting information

**S1 Appendix. Questionnaire.**
(DOCX)

**S2 Appendix. Interview Questions.**
(DOCX)

**S3 Appendix. Dataset.**
(XLSX)

## Author contributions

**Conceptualization:** Wanrong Lei, Zhibin Lei.

**Data curation:** Wanrong Lei.

**Investigation:** Zhibin Lei.

**Methodology:** Wanrong Lei.

**Project administration:** Wanrong Lei.

**Resources:** Wanrong Lei.

**Validation:** Wanrong Lei.

**Writing – original draft:** Zhibin Lei.

**Writing – review & editing:** Wanrong Lei.

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
