## [Decision Letter · Decision Letter 0]

22 May 2025

PONE-D-25-10820Exploring EFL Teachers’ Beliefs and Practices of Formative Assessment in Chinese ContextPLOS ONE

Dear Dr. Lei,

Thank you for submitting your manuscript to PLOS ONE. After careful consideration, we feel that it has merit but does not fully meet PLOS ONE’s publication criteria as it currently stands. Therefore, we invite you to submit a revised version of the manuscript that addresses the points raised during the review process.

We look forward to receiving your revised manuscript.

Kind regards,

Muhammad Zammad Aslam

Academic Editor

PLOS ONE

2. You indicated that ethical approval was not necessary for your study. We understand that the framework for ethical oversight requirements for studies of this type may differ depending on the setting and we would appreciate some further clarification regarding your research. Could you please provide further details on why your study is exempt from the need for approval and confirmation from your institutional review board or research ethics committee (e.g., in the form of a letter or email correspondence) that ethics review was not necessary for this study? Please include a copy of the correspondence as an ""Other"" file.

3. In the ethics statement in the Methods, you have specified that verbal consent was obtained. Please provide additional details regarding how this consent was documented and witnessed, and state whether this was approved by the IRB.

5. We note that your Data Availability Statement is currently as follows: [All relevant data are within the manuscript and its Supporting Information files.]

Additional Editor Comments:

Please follow the reviewers’ critiques, specifically relevant to self-reported data issues which also lack evidence from literature. I also outlined some issues that should be addressed alongside.

Issues to be addressed.

1. Generalizability of the Sample

- A reviewer mentioned that your data is confined to a specific geographical region. You should address this limitation in your ‘limitations’ section by arguing that your study can be expanded in future research through broader sampling and focused FA practices. Additionally, discuss these points in the future implications section. Furthermore, revise your ‘Discussion’ and ‘Conclusion’ sections accordingly.

2. Methodological Limitations

- According to the nature of your study, self-reported data may introduce bias, which could be clarified by expanding your methodology and focusing on future plans. This should also be addressed in your discussion while citing previous studies in the context of educational research.

3. Clarity on the process of participants’ recruitment

- Follow the comments of Reviewer 2 regarding this issue by clarifying the participants’ details about their experiences, session times, etc. Explain the consent obtained. Explain how you maintained the validity of the electronic survey.

4. Quantitative Reporting of Qualitative Data

- Please incorporate direct quotes in your qualitative analysis, properly citing participants wherever necessary, in accordance with thematic analysis standards. Update your tables accordingly. For example, there are issues with Table 8 and its interpretations, ensuring all quantitative data in the text aligns with the tables.

5. Discussion needs to be strengthened

- As discussed earlier, your discussion should correlate with your theoretical framework. Various discussed variables are not compared with prior literature, i.e., ‘feedback literacy’ is not discussed compared with the results of Carless (2020), and so on. Moreover, your gaps are also not clear in your discussions

6. More Important Suggestions

a. Include a ‘limitations’ section following the ‘discussion’ to address the previously mentioned limitations.

b. Improve clarity and readability by presenting a ‘visual model’ of FA.

c. Importantly, emphasize the ‘policy implications’ within the context of Chinese EFL teachers.

Reviewers' comments:

Reviewer's Responses to Questions

**Comments to the Author**

1. Is the manuscript technically sound, and do the data support the conclusions?

Reviewer #1: Partly

Reviewer #2: Yes

2. Has the statistical analysis been performed appropriately and rigorously? 

Reviewer #1: Yes

Reviewer #2: Yes

3. Have the authors made all data underlying the findings in their manuscript fully available?

Reviewer #1: Yes

Reviewer #2: Yes

4. Is the manuscript presented in an intelligible fashion and written in standard English?

Reviewer #1: Yes

Reviewer #2: Yes

5. Review Comments to the Author

Reviewer #1: The study’s sample is limited to university EFL teachers from a single province in China. While the sample includes a range of institutions and teacher demographics, it does not capture the full diversity of educational settings across China, such as variations in regional educational policies, resource allocation, or institutional cultures. This limits the generalizability of the findings to other provinces or to the national context, where formative assessment practices and challenges may differ significantly due to local factors, differing levels of economic development, or institutional priorities.

The reliance on self-reported data through questionnaires and interviews introduces the risk of response bias. Teachers may overstate their use of formative assessment or provide socially desirable responses, especially given the policy emphasis on formative assessment in China. Self-reporting also makes it difficult to verify whether stated practices align with actual classroom behaviors, potentially leading to discrepancies between reported and enacted formative assessment practices.

The study does not include direct classroom observations or analysis of formative assessment artifacts (such as student work samples, feedback records, or assessment rubrics). Without observational or documentary evidence, it is challenging to fully understand how formative assessment is implemented in practice, the quality of feedback provided, or the real-time challenges teachers face. This methodological limitation restricts the depth and validity of the findings regarding actual classroom practices. This should be discussed at length in the Discussion.

The manuscript is presented in an intelligible fashion and is written in standard English. The structure follows conventional academic organization, with clear sections for the abstract, introduction, literature review, methodology, findings, discussion, and conclusion. Each section logically builds on the previous one, and the arguments are supported by data and referenced literature, ensuring clarity and coherence throughout the text. Tables and figures are well-handled and referenced in the text, aiding reader comprehension. The manuscript also provides sufficient methodological detail for reproducibility and transparency, further contributing to its intelligibility.

Reviewer #2: The study presents a valuable and relevant investigation into formative assessment in EFL contexts and is generally well written and clearly structured. The topic is of practical importance and the mixed methods approach strengthens the research. However, several areas require clarification and improvement before the manuscript can be considered for publication.

1. It is unclear how two researchers managed to recruit and brief 322 participants within 60–90 days. It is recommended that the authors should clarify the recruitment process, including how many conferences were involved and how participants were approached and informed about the study.

2. I could not find page numbering, so I will be referring to the particular pages through page numbers as shown in the PDF file page count.

3. On pages 20, 21, and 23 (as per the PDF file), the authors report how many participants defined formative assessment in a particular way. Since qualitative data were thematically analyzed, reporting exact participant counts without context seems inconsistent with qualitative methodology. The authors should clarify their rationale for quantifying these themes or revise this reporting approach.

4. On page 28(as per the PDF file), the authors state that “36% of teachers report that formative assessment results...” This data point appears to be missing from Table 8. Please include this information in the table or clarify why it was excluded.

5. The discussion section currently reads more like an extension of the findings, with heavy use of quantitative data and limited integration of prior research. The authors are encouraged to engage more critically with existing literature, particularly studies on formative assessment in EFL contexts, to strengthen the scholarly grounding of their discussion.

6. The manuscript should be revised to fully comply with the Vancouver referencing style, as required by PLOS ONE. This includes proper in-text citations and reference formatting consistent with the journal standards.

The article has merit and makes a meaningful contribution but needs clarification and refinement in a few key areas before publication.

6. PLOS authors have the option to publish the peer review history of their article (what does this mean? ). If published, this will include your full peer review and any attached files.

**Do you want your identity to be public for this peer review?** For information about this choice, including consent withdrawal, please see our Privacy Policy .

Reviewer #1: No

Reviewer #2: No

---

## [Author Response · Author response to Decision Letter 1]

19 Jun 2025

Dear Editor and Reviewers,

Thank you for your thorough review and constructive feedback on our manuscript. Please find our detailed point-by-point response in the attached “Response to Reviewers” file. Below is a summary of the key changes we have made:

1. Funding Information and Financial Disclosure Consistency

Comment: The grant information provided in the “Funding Information” and “Financial Disclosure” sections do not match.

Response: We have confirmed that the grant numbers and funders listed in the “Funding Information” section are correct. We apologize for the oversight in filling the “Financial Disclosure” section previously (Location: Funding Information Section).

2. Data Availability Statement

Comment: Confirm whether the submission contains all raw data required to replicate the results.

Response: We have included a minimal data set required for replication, attached as “dataset” in Supporting Information (Location: S4).

3. Supporting Information Captions

Comment: Include captions for Supporting Information files at the end of the manuscript.

Response: Captions for all Supporting Information files (S1, S2, S3, S4) have been added at the end of the manuscript. In-text citations have been updated accordingly (Location: End of Manuscript).

4. Methodological Limitations

Comment: Address potential bias from self-reported data and expand methodology.

Response: We have expanded the methodology section to discuss potential biases and future plans. This includes adding a new “Ethical Considerations” section (Location: Section 3.5 Ethical Considerations).

5. Generalizability of the Sample

Comment: Discuss geographical limitations and future implications.

Response: We have added a “Limitations” section addressing geographical limitations and discussed expansion opportunities in the future implications section (Location: Section 6 Limitations, P. 44-45).

6. Discussion Section

Comment: Engage more critically with existing literature, particularly studies on formative assessment in EFL contexts.

Response: We have revised the “Discussion” section to deepen engagement with existing literature, particularly focusing on studies by Carless & Winstone (2020), Xu & Brown (2016), and others (Location: Sections 5.1–5.3).

7. Vancouver Referencing Style Compliance:

Comment: Ensure compliance with Vancouver referencing style.

Response: The manuscript has been revised to fully conform to the Vancouver referencing style, including in-text citations and reference list formatting (Location: Entire text and References).

We are grateful for your continued support and consideration. Should there be any further questions or if additional modifications are needed, we are more than happy to make any further adjustments.

Thank you again for your time and consideration.

Sincerely

Wanrong Lei

---

## [Decision Letter · Decision Letter 1]

6 Aug 2025

PONE-D-25-10820R1Exploring EFL Teachers’ Beliefs and Practices of Formative Assessment in Chinese ContextPLOS ONE

Dear Dr. Lei,

Thank you for submitting your manuscript to PLOS ONE. After careful consideration, we feel that it has merit but does not fully meet PLOS ONE’s publication criteria as it currently stands. Therefore, we invite you to submit a revised version of the manuscript that addresses the points raised during the review process. Please submit your revised manuscript by Sep 20 2025 11:59PM. If you will need more time than this to complete your revisions, please reply to this message or contact the journal office at plosone@plos.org . Please include the following items when submitting your revised manuscript:

We look forward to receiving your revised manuscript.

Kind regards,

Muhammad Zammad Aslam, Ph.D.

Academic Editor

PLOS ONE

Journal Requirements:

Additional Editor Comments:

The manuscript needs to fulfill the basic requirements following the traditional guidelines, such as explanations of teaching intervention for replication of the present study (such as detailed curriculum, description of texts or methods used, or other supporting educational material. If materials, methods, and protocols are well established, authors may cite articles where those protocols are described in detail, but the submission should include sufficient information to be understood independent of these references.

Positive Elements:

a) Evaluation of Sufficiency of Materials and Methods

The manuscript provides reasonable detail about both teaching approaches:

1. Lecturer-Guided Group; the intervention included a personal narrative from a guest lecturer with lived experience of depression. Topics covered included:

- Triggers of depression

- Recovery process

- Daily life during treatment

- Support systems

- Introduction to counselling services

- Personal message to students

Moreover, the background of the lecturer (age, past job, diagnosis timeline) is also explained in rich detail.

2. Non-Lecturer-Guided Group: A Health education teacher who is experienced in the field taught the necessary topics; Psychosomatic correlations, Seeking help, and maintaining social support

3. Curriculum & Tools:

- The five domains are assessed through a 24-item questionnaire, which is also validated by experts

- Folowchart for the Lesson plan is provided (pre/post/follow-up survey)

- Supporting materials (datasets, questionnaire in Japanese, ethics approvals, etc.) are present.

- A video follow-up (3 months later)

However, some areas should need enhancements clarifying objectively:

1. Lesson Plan Replicability:

- A full transcript may be outlined alongside the lecturer's session, alongside the narrative structure. Including an example lesson plan or structured summary would support replicability.

- Similarly, names of the chapters, names of the textbooks, or any other instructional materials may be needed alongside the teacher-led textbook content descriptions.

- The details about the videos, that is included in follow-up classes, should be needed. For instance, who is the creator? What is the script? How is this script relevant? Is the duration of the video is sufficient? What is the source of the video? etc

2. Availability of Teaching Materials:

- Please provide an English version of the questionnaire, which is a supplement for the international or multidisciplinary audiences.

- Are the lectures' notes, scripts, and slides available on request?

In summary, areas for improvement are:

- The description of lesson content, particularly for the teacher-led session, would benefit from the inclusion of exact textbook references, lesson objectives, or example discussion questions.

- For the lecturer-guided session, while the biographical narrative is informative, a more structured or script-based summary (e.g., main talking points, message themes) would be helpful for educators attempting to replicate the intervention.

- The video resources used in follow-up classes are not described in sufficient detail (e.g., content summary, creator, duration).

- The authors may consider providing an English version of the questionnaire in supplementary material for international audiences.

Educators and researchers, who are likely to adapt the intervention in different contexts, would benefit from the above changes.

Reviewers' comments:

Reviewer's Responses to Questions

**Comments to the Author**

1. If the authors have adequately addressed your comments raised in a previous round of review and you feel that this manuscript is now acceptable for publication, you may indicate that here to bypass the “Comments to the Author” section, enter your conflict of interest statement in the “Confidential to Editor” section, and submit your "Accept" recommendation.

Reviewer #1: All comments have been addressed

Reviewer #2: All comments have been addressed

2. Is the manuscript technically sound, and do the data support the conclusions?

Reviewer #1: Yes

Reviewer #2: Yes

3. Has the statistical analysis been performed appropriately and rigorously? 

Reviewer #1: Yes

Reviewer #2: Yes

4. Have the authors made all data underlying the findings in their manuscript fully available?

Reviewer #1: Yes

Reviewer #2: Yes

5. Is the manuscript presented in an intelligible fashion and written in standard English?

Reviewer #1: Yes

Reviewer #2: Yes

6. Review Comments to the Author

Reviewer #1: Thank you very much for addressing my suggestions; I see that they have all been correctly incorporated.

Reviewer #2: I have read the author's response and deeply appreciate the effort and patience taken to address my comments. I look forward to seeing this research published.

7. PLOS authors have the option to publish the peer review history of their article (what does this mean? ). If published, this will include your full peer review and any attached files.

**Do you want your identity to be public for this peer review?** For information about this choice, including consent withdrawal, please see our Privacy Policy .

Reviewer #1: No

Reviewer #2: No

---

## [Author Response · Author response to Decision Letter 2]

9 Sep 2025

Dear Editor,

Thank you for your careful review and the opportunity to revise our manuscript. We sincerely appreciate the time and thoughtful feedback from both the reviewers and the editorial team.

We are grateful that Reviewer #1 and Reviewer #2 have confirmed that all previous comments have been adequately addressed, and that they find the manuscript technically sound, well-analyzed, data-available, and clearly written.

Regarding your editorial comments, we would like to clarify a key point: This study is not an intervention study, but a mixed-methods investigation into EFL teachers’ beliefs and practices regarding formative assessment in the Chinese context. Therefore, there was no teaching intervention, no lesson plans, no video materials, and no instructional scripts developed or implemented as part of this research.

Our data were collected through a survey and semi-structured interviews with in-service EFL teachers. Accordingly,

1.There are no lesson plans, textbooks, or video materials to provide, as none were used in the study.

2.The English version questionnaire has been included in the Supporting Information compressed archive (“supporting materials.zip”`) as “S1_questionnaire.docx” in the system and remains accessible in this revision.

3. No lecture notes, scripts, or slides were generated, as this was not an intervention-based study.

To improve clarity, we have:

Revised the Abstract to more accurately reflect the qualitative, non-interventional nature of the study and the data collection methods.

We hope these clarifications address your concerns. We hope the revised manuscript now clearly communicates the nature of the study and meets the standards of PLOS ONE.

Thank you again for your guidance.

Sincerely,

Dr. Wanrong Lei

---

## [Editor Report · Decision Letter 2]

17 Sep 2025

Exploring EFL teachers' beliefs and practices of formative assessment in Chinese context

PONE-D-25-10820R2

Dear Dr. Lei,

We’re pleased to inform you that your manuscript has been judged scientifically suitable for publication and will be formally accepted for publication once it meets all outstanding technical requirements.

Kind regards,

Muhammad Zammad Aslam, Ph.D.

Academic Editor

PLOS ONE
---

## [Editor Report · Acceptance letter]

PONE-D-25-10820R2

PLOS ONE

Dear Dr. Lei,

I'm pleased to inform you that your manuscript has been deemed suitable for publication in PLOS ONE. Congratulations! Your manuscript is now being handed over to our production team.

Kind regards,

on behalf of

Dr. Muhammad Zammad Aslam

Academic Editor

PLOS ONE